# Emerging Bluetongue Virus Serotype 4 in the Balearic Islands, Spain (2021): Outbreak Investigations and Experimental Infection in Sheep

**DOI:** 10.3390/microorganisms13020411

**Published:** 2025-02-13

**Authors:** David Romero-Trancón, Marta Valero-Lorenzo, María José Ruano, Paloma Fernández-Pacheco, Elena García-Villacieros, Cristina Tena-Tomás, Ana López-Herranz, Jorge Morales, Bartolomé Martí, Miguel Ángel Jiménez-Clavero, Germán Cáceres-Garrido, Montserrat Agüero, Rubén Villalba

**Affiliations:** 1Laboratorio Central de Veterinaria (LCV), Ministry of Agriculture, Fisheries and Food (MAPA), 28110 Algete, Spain; davidromerotrancon@gmail.com (D.R.-T.); mvalero@mapa.es (M.V.-L.); mruanor@mapa.es (M.J.R.); alherranz@mapa.es (A.L.-H.); jmbello@mapa.es (J.M.); rvillalba@mapa.es (R.V.); 2Centro de Investigación en Sanidad Animal (CISA-INIA), Consejo Superior de Investigaciones Científicas (CSIC), 28130 Valdeolmos, Spain; pacheco@inia.csic.es (P.F.-P.); majimenez@inia.csic.es (M.Á.J.-C.); 3Subdirección General de Sanidad e Higiene Animal y Trazabilidad, Ministry of Agriculture, Fisheries and Food (MAPA), 28014 Madrid, Spain; egvillacieros@mapa.es (E.G.-V.); gcaceres@mapa.es (G.C.-G.); 4Tecnologías y Servicios Agrarios, S.A. (TRAGSATEC), 28037 Madrid, Spain; at_algete9@mapa.es; 5Laboratorio de Sanidad Animal del Instituto de Investigación y Formación Agroalimentaria y Pesquera de las Islas Baleares (IRFAP), Conselleria d’Agricultura, Pesca i Medi Natural de les Illes Balears, 07009 Palma, Spain; tmarti@irfap.es

**Keywords:** bluetongue virus, experimental infection, diagnosis, emerging virus, serotyping, sheep, outbreak, epizootic

## Abstract

Bluetongue is an infectious, non-contagious, arthropod-borne viral disease of ruminants caused by bluetongue virus (BTV), producing severe impacts on livestock. Historically, Southern Europe has suffered multiple incursions of different BTV serotypes with serious consequences. In 2021, BTV re-emerged in the Balearic Islands (Spain) after 16 years free of the disease, causing a large outbreak that mainly affected sheep, as well as cattle and goats. A novel emerging strain of BTV serotype 4 (BTV-4) was identified via preliminary molecular characterization as the etiological culprit of the epizootic. This study delineates the outbreak in the Balearic Islands in 2021, encompassing field-based clinical observations and laboratory findings. Additionally, an experimental infection was conducted in sheep using the novel BTV-4 strain to assess its virulence, pathogenicity, and laboratory diagnostic characteristics. The in vivo characterization was conducted concurrently with the virulent and widely disseminated BTV-4 RNM 2020 strain that has precipitated significant outbreaks in the Mediterranean region in recent years. Both strains exhibited analogous pathogenic potential in sheep and yielded equivalent outcomes in diagnostic parameters. Furthermore, the impact of the novel BTV-4 strain is discussed.

## 1. Introduction

Bluetongue virus (BTV) is an arthropod-borne pathogen transmitted by *Culicoides* spp. biting midges that affects domestic and wild ruminants [1,2]. BTV infection causes the bluetongue disease, which can lead to high morbidity and mortality mainly in sheep and less frequently in cattle, goats, and camelids, which usually remain asymptomatic, acting as reservoirs [3,4,5,6,7]. The virus is classified within the *Orbivirus* genus (*Sedoreoviridae* family) and possesses a characteristic genome consisting of 10 linear double-stranded RNA segments, which facilitates significant genetic diversity and the emergence of reassortant strains [8,9,10]. BTV is classified into serotypes based on segment 2, encoding the main neutralization antigen within the VP2 structural protein [11,12]. Currently, 36 serotypes have been described [13], of which 24 (BTV-1 to -24) are recognized as classical serotypes [2]. Classical serotypes have the potential to produce the disease [14] and largely fail to confer cross-protective immunity [15,16]. The pathogenesis of bluetongue has been well characterized in sheep, including reflect congestion, edema, and hemorrhage as a consequence of virus-mediated vascular injury [17]. This leads to the production of a combination of fever, serous or bloody nasal discharge, respiratory distress, cyanoses in the tongue, oral erosions and ulcers, lameness with coronitis, weakness, and death [10]. Because of the severe consequences in livestock and in the economy, bluetongue is listed as a multispecies disease by the World Organization for Animal Health (WOAH) and categorized as a C + D + E disease in the European Union, being classical serotypes subjected to compulsory notification [18,19].

The incidence of the bluetongue in Europe was initiated primarily in the Mediterranean basin, where BTV has produced periodic incursions [20,21]. The first records of BTV in Europe are referred to outbreaks in Cyprus in 1924 (BTV-4) [22], followed by sporadic incursions in the Iberian Peninsula in 1956 and 1960 (BTV-10) [23] and subsequent outbreaks in the Greek islands of Lesbos in 1979 and Rhodes in 1980 (BTV-4 in both) [20]. The incidence and distribution in Europe changed dramatically in 1998, when BTV-9 was detected in several Greek islands and spread through mainland Greece and the Balkan Peninsula [24]. In the following years, several incursions of multiple serotypes caused the distribution of the virus into naive regions in Europe and increased the diversity of viral strains in endemic areas across the continent [25,26]. Specifically, from 1998 to 2006, BTV-1, -2, -4, -9, and -16 spread throughout the European region of the Mediterranean coast [27]. However, the paradigm changed in 2006, when the emergence of a BTV-8 strain of sub-Saharan origin in the Netherlands [28] caused a severe epizootic that extended the distribution area of the virus as far north as 53° N in Northern Europe [29]. Afterwards, several European regions suffered the emergence and re-emergence of different BTV classical strains (BTV-3, -4, -11, -12, and -14) threatening the continent [30,31,32,33,34].

It is worth highlighting the incursion and re-emergence of different strains of BTV-4 and their expansion throughout Mediterranean coastal regions. Until 2014, two main BTV-4 phylogenetic groups were identified in the Mediterranean basin [35]. The first group in the Eastern Mediterranean region is related to Greek strains isolated in 1979, 1999, and 2000 (Eastern topotype); and a second one was detected in 2004 in the Western Mediterranean region, with strains detected in Corsica, Morocco, Tunisia, Algeria, Spain, and Italy linked to the wind-driven dissemination of infected midges from North Africa (Western topotype) [33,35]. In 2014, a novel BTV-4 strain not related to those Mediterranean phylogenetic groups was detected in Greece and rapidly spread across the Balkan Peninsula [33]. This strain has subsequently been detected in neighboring areas, causing large outbreaks, including Western Mediterranean regions such as Italy, Corsica, and Mainland France [33,36,37], and re-emerging in the Balkans in 2020 [38,39]. In general, BTV incursions in Europe have been related to viral strains originally from North Africa—invading primarily Southern and Western European regions—and to the dispersal of viral strains from the Near and Middle East into the Balkan peninsula [33,35,38,40].

Particularly in Spain, several outbreaks by different BTV serotypes have been reported. In 1956, a BTV-10 strain related to North African strains was introduced in the Iberian Peninsula, affecting Portugal and Spain and causing a severe epizootic [41,42]. This event was eradicated in 1958 [43], and no new outbreaks were detected in Spain until October 2000, when a BTV-2 strain related to Sub-Saharan strains was detected in the Balearic Islands [44]. BTV-2 did not spread from the Balearic Islands and was eradicated in 2003. Since then, incursions of other serotypes have been reported, first in 2003 in the Balearic Island and then in 2004 in mainland Spain, both with a BTV-4 strain detected alongside in Morocco and Portugal that spread throughout Western Mediterranean areas in the following years [35,40]. Later, in 2007, BTV-1 was detected in Southern Spain, and in 2008, the BTV-8 strain that had been circulating in Northern and Central Europe since 2006 appeared in the north of the Iberian Peninsula [23]. Recently, the BTV-3 strain from Northern Europe and Portugal has also been introduced in mainland Spain [32,45,46]. Different measures, including massive vaccinations, have been adopted for years to control and eradicate the disease [47].

In June 2021, BTV-4 was detected again in the Balearic Islands after 16 years of being officially free from the disease. The Official Veterinary Services notified a BTV serological suspicion in a sentinel bovine herd on the island of Majorca and later reported clinical suspicions in sheep herds in the same area. The Laboratorio Central de Veterinaria based in Algete (Madrid), as the National Reference Laboratory (NRL) in Spain for bluetongue [48], identified and confirmed BTV-4 according to the reference diagnostic methods in place. After the first detections, a restricted area was established around the outbreak, which included all the Islands of the Balearic archipelago (Majorca, Minorca, Ibiza, Cabrera, and Formentera), where prevention, surveillance, and control measures were reinforced. In mid-July 2021, a mass vaccination program was implemented, bringing the situation under control by the end of the year.

This research details the field-based clinical observations and laboratory findings during the outbreak, including genetic characterization of the viral isolate via partial genome sequencing of segments 2, 5, and 10. Further, a controlled experimental infection was conducted to evaluate the virulence, pathogenicity, and laboratory diagnostic features of the novel strain. The findings are discussed with a focus on the epidemiology and pathogenicity of BTV strains in the Mediterranean region, as well as on the crucial importance of laboratory diagnostic activities.

## 2. Materials and Methods

### 2.1. Diagnosis of Bluetongue Outbreak in the Balearic Island

#### 2.1.1. Sampling by the Official Veterinary Services

In the framework of the bluetongue surveillance, control, and eradication program [47] in force in Spain since 2004, serum and EDTA blood samples from a sentinel bovine, which was ELISA positive in the regional official laboratory, were received at the end of June 2021 in the NRL. Later, after confirming the index case, 427 serum and 777 EDTA-blood samples were received in the NRL from June to mid-November from 280 clinical or serological suspicious farms, 274 of them sheep, 4 bovine, and 2 goat farms in Majorca, Minorca, or Ibiza islands (Appendix A). Additionally, to evaluate the immune response after vaccination with a BTV-4 inactivated vaccine, EDTA blood and serum samples from 62 vaccinated sheep in non-affected areas were taken 37 to 62 days after vaccination. Serum samples were analyzed using serological tests and EDTA blood samples via virological diagnosis, as described below.

#### 2.1.2. Serological Test Flow Chart

Serum samples were stored at 4 °C until analyzed firstly via the blocking ELISA (b-ELISA) kit INgezim BTV Compac 2.0 (Ingenasa, Madrid, Spain), and negative or doubtful samples were analyzed with a second double recognition ELISA (dr-ELISA), INgezim BTV DR (Ingenasa, Madrid, Spain). ELISA-positive samples in one of the ELISA test were serotyped via virus neutralization test (VNT) using the three BTV serotypes that had recently circulated in Spain at that moment (BTV-1, -4, and -8). Details on these methods are included in Section 2.3.

#### 2.1.3. Virological Diagnosis Flow Chart

Extracted nucleic acids from EDTA blood samples were subjected firstly to a serogroup-specific (GS) real time reverse transcriptase polymerase chain reaction (rRT-PCR) that targets segment-10 of the viral genome. Positive samples were subsequently typed using serotype-specific (TS) rRT-PCR methods targeting segment-2; they were developed and validated to specifically detect the BTV-1, -4, and -8 strains that historically have circulated in Spain and neighboring countries. Two rRT-PCR-positive EDTA blood samples from farm 2 (Appendix A) were selected to carry out virus isolation and subsequently partial sequencing of the viral genome. Details on these methods are included in Section 2.3.

### 2.2. Experimental Infection of BTV-4 in Sheep

#### 2.2.1. Study Design

Twelve crossbred sheep were selected for the trial, aged approximately 12–24 months. The animals were housed in two groups of six randomly selected sheep in two independent boxes at the high biocontainment animal facility of the CISA-INIA, CSIC (Valdeolmos, Madrid, Spain). All animals had been previously tested using ELISA and GS rRT-PCR to discard a BTV infection. After one week of acclimatization, five sheep from each group were challenged with a widespread BTV-4 strain (Group 1: BTV4 RNM 2020) or with the BTV-4 strain isolated from the Balearic Islands outbreak (Group 2: BTV-4 SPA (BAL) 2021). One animal in each group was kept sham-inoculated as a non-infection control (Figure 1).

#### 2.2.2. Inoculum and Inoculation of Animals

Both BTV-4 strains were isolated from EDTA blood samples received from the respective outbreaks: BTV-4 outbreak in 2020 in Republic of North Macedonia and BTV-4 outbreak in 2021 in the Balearic Islands. Both inocula, BTV-4 RNM 2020 and BTV-4 SPA (BAL) 2021, were prepared after the same number of passages (1 in KC cells plus 3 in BHK cells plus 1 again in KC cells) and obtained by recovering the supernatant of the infected KC cell culture after seven days. Three ml per animal of infectious virus at a concentration of 10^5^ TCID_50_/mL were inoculated subcutaneously in two inoculation points in the axillary regions. In addition, the presence of BTV-4 was confirmed in each inoculum via GS and BTV-4 TS rRT-PCR before inoculation. The non-infected control animal of each group was inoculated in the same way with the supernatant of an uninoculated KC cell culture confirmed BTV-negative by GS rRT-PCR.

#### 2.2.3. Sampling and Clinical Monitoring of the Animals

Animals were monitored daily to detect the onset of clinical signs and to record rectal temperatures (considered as fever when >40 °C) until day 39 post-inoculation (dpi). Whole EDTA blood and serum samples were collected according to the scheme presented in Figure 1 to assess the presence of BTV genome (RT-PCRemia), infectious virus (viremia), and serological response. Randomly selected animals (one per group) were sequentially sacrificed at different dpi, as indicated in Figure 1, to determine the presence of gross lesions and virus tropism.

#### 2.2.4. Virological Analyses

EDTA blood samples were kept refrigerated at 4 °C after being obtained until analyzed for GS rRT-PCR. The tissue samples, consisting of 0.1 g from liver, heart, spleen, lung kidney, mesenteric, and mediastinal lymph nodes, added to PBS (1 mL), were homogenized using an automatic homogenizer with glass spheres. The homogenized tissues were then centrifuged at 1000× *g* for 15 min at 4 °C, and supernatants were stored at +4 °C until analyzed via GS rRT-PCR. Finally, to determine the presence of infectious virus, rRT-PCR-positive samples were subsequently analyzed via virus isolation. Description of virological methods is presented in Section 2.3.

#### 2.2.5. Analyses of Serological Response

Serum samples were analyzed to detect the presence of antibodies against the serogroup-specific VP7 protein using the b-ELISA kit INGEZIM BTV Compac 2.0 (Ingenasa, Madrid, Spain). In addition, the presence of neutralizing antibodies against BTV-4 was analyzed via VNT in ELISA-positive sera. Description of both serological methods is indicated in Section 2.3.

#### 2.2.6. Statistical Analysis

Statistical significance of the observed differences in rectal temperatures, BTV-RNA detection in blood via rRT-PCR, and antibody detection with ELISA and VNT titers were evaluated using the non-parametric Mann–Whitney U test. *p* values equal to or less than 0.05 were considered statistically significant.

### 2.3. Diagnostic Methods

#### 2.3.1. ELISA Analysis

Both blocking and double recognition ELISA assays were performed following the manufacturer’s protocols, and Optical Density (OD) was measured at 450 nm. Samples analyzed with INgezim BTV Compac 2.0 were considered positive if their blocking percentage [100 − (sample OD × 100/Negative control OD)] was equal or over the cut-off value set at 40%, negative if that percentage was equal or lower than 35%, and doubtful if their blocking percentage was between both cut-off values. Samples analyzed with INgezim BTV DR were considered positive if the OD value at 450 nm was higher than the cut-off value established for each assay according to the instructions provided by the manufacturer (15% of positive control).

#### 2.3.2. Serotyping via Virus Neutralization Test (VNT)

Serotype-specific neutralizing antibodies were detected according to a VNT method following the EURL protocol [49] based on the WOAH Manual [50]. Briefly, two-fold dilutions of sera, starting at 1:5 dilution, were challenged with the BTV serotype object of study. Titers were expressed as the reciprocal of the highest dilution of serum able to neutralize 100 TCID_50_ of the corresponding BTV serotype. Analyzed samples were considered as positive if ≥1:5, and the final titer was determined according to the Spearman–Kärber method [51,52]. BTV strains used for VNT were BTV1 ALG 2006/01, BTV-4 SPA 01/2004, and BTV8 BEL 2006/01.

#### 2.3.3. Nucleic Acid Extraction

For molecular diagnosis, nucleic acid extraction was performed from 200 µL of EDTA-blood, homogenized tissue, or virus suspension with the commercial IndiMag Pathogen Kit (Indical Bioscience, Leipzig, Germany) in a BioSprint 96 automated extraction system (Qiagen, Hilden, Germany) according to the manufacturer’s instructions. Nucleic acid was eluted in a final volume of 100 µL of nuclease-free water and kept at −80 °C until testing with molecular methods.

#### 2.3.4. BTV Genome Detection by Serogroup and Serotype Specific Real Time RT-PCR Methods

For the GS, rRT-PCR detection used the method described in the WOAH Manual [50] and developed by Hofmann et al. (2008) [53], which targets segment-10 of the viral genome. For typing, TS rRT-PCR methods targeting segment-2 were developed and validated to specifically detect the BTV-1, -4, and -8 strains that historically have circulated in Spain and neighboring countries. The primers, probes, amplicon sizes, and strains for which these methods were validated are shown in Table 1.

GS and TS rRT-PCR protocols were similar, with slight differences in primer concentration and PCR reaction program. Each rRT-PCR assay was performed in a final volume of 20 µL using the commercial kit known as AgPath-ID™ One-Step RT-PCR Reagents (Applied BioSystems, Whaltman, MA, USA). Briefly, 2 µL of isolated RNA was mixed with the corresponding forward and reverse primers in ribonuclease (RNase)-free water up to 7 µL (primer concentration: GS 3.57 µM or TS 2.86 µM; final concentration: GS 1.25 µM or TS 1.00 µM). This mixture was denatured by heating at 95 °C for 5 min, followed by rapid cooling on ice. Then, a mixture of AgPath-ID™ One-Step RT-PCR Reagents (Applied BioSystems, Whaltman, MA, USA) and the corresponding probe was added per reaction well, reaching a final concentration probe of 0.25 µM. Reactions were performed in the 7500 fast real-time PCR systems (Applied Biosystems), and the rRT-PCR data were analyzed with the 7500 software, version 2.3 (Applied Biosystems). The rRT-PCR reaction program was 10 min at 48 °C and 10 min at 95 °C, followed by 40 cycles of 2 s at 97 °C and 30 s at 56 °C (for the GS method) or 60 °C (for the TS methods). Samples were considered positive when a typical amplification curve was obtained, and the cycle threshold (Ct) value was lower or equal to 35 (Ct ≤ 35), inconclusive when Ct > 35, and negative when no curve was obtained.

#### 2.3.5. Virus Isolation in Cell Culture

EDTA blood samples (1 mL) were centrifuged at 1000× *g*/10 min/4 °C, the supernatants were discarded, and the cell pellets were washed three times with phosphate-buffered saline (PBS). Next, the erythrocytes were lysed via osmotic shock by adding 1 mL of sterile distilled water and maintaining the vial in ice for 10 min. Finally, after centrifugation at 12,000× *g*/5 min/4 °C, the supernatants were removed, 0.5 mL of PBS was added to the cell debris pellets, and the samples were stored at 4 °C until used.

The homogenized tissue supernatants were passed through a 0.45 µm pore filter, and 1% of Antibiotic-Antimycotic solution (stabilized with 10.000 units of Penicillin, 10 mg of Streptomycin, and 25 µg of Amphotericin B per ml, sterile-filtered) was added and incubated for 20 min at room temperature. Finally, samples were stored at −80 °C until used.

Virus isolation was performed following a standardized protocol for BTV isolation [49]. Firstly, 0.2 mL of sample (washed and lysed EDTA-blood diluted in PBS in a 1:4 proportion or homogenized tissue) was inoculated on 24-well plates containing confluent monolayers of KC cells (derived from *Culicoides sonorensis*) prepared the day before. After 1 h of incubation at 28 °C, 0.8 mL of Schneider’s Drosophila Medium (enriched with 1% antibiotic/antimycotic, 1% L-Glutamine, 1% non-essential amino acids, and 5% fetal bovine serum heat inactivated at 56 °C/1 h) was added and incubated for 7 days in the same conditions. Since KC cell line does not usually show cytopathic effect (CPE), 0.2 mL of the resuspended KC cell culture was inoculated in a monolayer of BHK-21 cells in 24-well plates. After 1 h of incubation at 37 °C and 5% CO_2_ for virus absorption, 0.8 mL of complete EMEM (supplemented with 1% L-Glutamine, 1% non-essential amino acid solution, 1% antibiotic/antimycotic solution, and 2% fetal bovine serum heat inactivated at 56 °C/1 h) was added and incubated in the same conditions. Inoculated BHK-21 cell cultures were observed daily for 5 days. When no signs of CPE appeared by day 5, 0.2 mL was inoculated in a fresh monolayer of BHK-21 cell culture in 24-well plates before considering them as negative. Viral isolation was achieved when CPE was evident, and comparison of the Ct value between consecutive passages, obtained via GS rRT-PCR, was compatible with the presence of virus replication.

#### 2.3.6. Partial Genome Sequencing and Sequencing Analysis

Gel-based RT-PCRs were performed using nucleic acid extracted from a BTV-4 SPA (BAL) 2021 viral suspension to partially amplify segments 2, 5, and 10 of the virus according to previously described methods [54,55,56]. The PCR products were visualized in a 2% horizontal electrophoresis agarose gel, purified using the QIAquick^®^ PCR Purification Kit (Qiagen, Hilden, Germany), and sequenced by Sanger technology based on the ABI Prism BigDye Terminator v3.1 Cycle sequencing kit on a 3730 Genetic Analyser (Applied Biosystems, Foster City, CA, USA). Nucleotide sequences were analyzed and assembled into consensus sequences using the DNA Sequencing Analysis Software Version 6.0 and SeqScape Version 3.0 (Applied Biosystems, Foster City, CA, USA). Obtained sequences were then compared with the homologous segments accessible in GenBank.

## 3. Results

### 3.1. Timeline and Progression of the Epizootic in the Balearic Islands

On the 15th of June 2021, a positive b-ELISA result in an asymptomatic sentinel bovine was reported in the municipality of Pollença (Majorca). The farm participated in the National surveillance program for bluetongue and had a census of 4 cattle and 19 sheep. On the 23rd of June, the NRL confirmed the positive result as BTV-4 via serological and molecular methods. One week later (calendar week 26), nine clinical suspicions in sheep herds were also confirmed as BTV-4. Afterwards, the number of reports increased throughout the island of Majorca, with 10 new confirmed reports by calendar week 34 and 45 positive herds by calendar week 39. On the 8th of October, the virus spread to the island of Ibiza, and on the 27th of October, it was also detected on the island of Minorca, affecting a sheep farm. A peak in infections was detected in October, with 190 new confirmed infected herds in calendar weeks 40 to 43. The total number of farms affected was 283 (276 in Majorca, 6 in Ibiza, and 1 in Minorca) (Figure 2).

Except for the index case and three other cattle farms and one goat farm, the outbreaks were mainly reported on sheep farms. Based on the official information, the morbidity rate considering the total census in positive farms was 4.7% in sheep (1.5% mortality). Four cattle farms were confirmed positive with a global morbidity of 5% in cattle (no mortality). Only one goat flock was declared positive, with a within-herd morbidity of 11% (4.8% mortality). Most outbreaks were confirmed via GS rRT-PCR and typed in the NRL (Appendix A), but some of them were notified by the description of an epidemiological link together with the presence of clinical signs and a positive ELISA result in the regional laboratory.

After the outbreak was declared, a mass vaccination program was launched in the Balearic Islands from mid-July 2021 with the BLUEVAC BTV-4 vaccine (CZ Vaccines S.A.U., Pontevedra, Spain) in sheep and cattle. The last clinical suspicion was reported on the 26th of November 2021 (calendar week 46), bringing the outbreak under control by the end of the year. During the following two transmission seasons (2022 and 2023), no new positive cases were reported, and the European Union declared the area free of the disease in January 2024.

### 3.2. Clinical Signs Observed in Sheep, Cattle, and Goats in the Field During the Outbreak

Since the beginning of the outbreak, clinical signs were observed, mainly in sheep. BTV-4 infected sheep showed dyspnoea, serous nasal discharge, ptyalism, glossitis, hyperthermia, apathy, blepharitis, congestion of the oral and periocular mucosa, the presence of ulcers in the mouth, and facial and submandibular edema (Figure 3a–g). In cattle, two farms were notified by seroconversion in sentinel animals without clinical signs, and only four animals from the other three notified farms showed mild clinical signs (affected animals/farm census: 2/31, 1/14, 1/30), including hypersalivation and the presence of facial edemas (Figure 3h,i). In the infected goat herd, mild clinical signs were reported, affecting a total of 5 animals out of a census of 42 goats.

### 3.3. Laboratory Diagnosis

#### 3.3.1. Virological Diagnosis

A total of 558 out of 778 EDTA blood samples from 260 farms were confirmed positive via GS rRT-PCR (Appendix A) and subsequently typed as serotype 4 at the NRL, while no amplification was observed when using the typing TS rRT-PCR methods for BTV-1 and BTV-8. It is worth noting that the first 192 samples typed were analyzed with both BTV-4 typing methods (BTV-4 w.med and BTV-4 extended): while all samples were positive with the BTV-4 extended TS rRT-PCR method, only 25.52% (49/192) of the samples were positive, and 40.10% (77/192) were inconclusive using the BTV-4 w.med typing method. Furthermore, the Ct values of the 49 samples that were positive with the BTV-4 w.med typing method were, on average, 11 Cts higher compared to those obtained with the extended BTV-4 typing method (Appendix A).

Virus isolation was performed on the two EDTA blood rRT-PCR-positive samples from farm 2 (Appendix A), obtaining positive results. The viral isolate from sample 1 was partially sequenced and used for the experimental inoculation.

#### 3.3.2. Serological Diagnosis

A total of 428 serum samples were analyzed, comprising 263 from GS rRT-PCR-positive animals in affected farms and 165 from GS rRT-PCR-negative animals from both infected or non-infected farms (sampled from 10 to 90 days post-vaccination).

Following the NRL serological testing flow, 98% (258/263) of the infected animals (rRT-PCR-positive) and 76.36% (126/165) of rRT-PCR-negative animals presented antibodies against BTV by ELISA (Appendix A).

The majority of positive or doubtful samples in ELISA were then analyzed via VNT against serotypes BTV-1, -4, and -8, and 75.51% (182/241) of rRT-PCR-positive animals and 38.98% (46/118) of rRT-PCR-negative animals were positive against BTV-4, showing titers higher than 1:40 in 80% and 45% of VNT-positive samples, respectively. All the ELISA positive or doubtful samples were VNT-negative for BTV-1 and BTV-8 (Appendix A).

#### 3.3.3. Post-Vaccination Study

As for the 62 serum samples from vaccinated animals in non-affected areas, all paired EDTA blood samples were negative according to GS rRT-PCR, confirming the absence of ongoing infection. After applying the flow chart of b-ELISA and the dr-ELISA test, a positive or doubtful serological response was observed in 52 samples (83.8%). A total of 47 ELISA-positive or doubtful samples were serotyped via VNT against serotype BTV-4, and 19.14% (9/47) were positive, showing titers of 1:5–1:10 in seven animals and 1:40 in two of them (Appendix A).

#### 3.3.4. Partial Genome Sequencing

Partial sequences of viral segments 2, 5, and 10 were obtained for the BTV-4 virus isolated from sample 1 in farm 2 and named BTV-4 SPA (BAL) 2021. The comparison with the homologous segments accessible in GenBank showed the highest similarity (>99.7% in the three analyzed segments) with BTV-4/21-03 CORSICA 2021, a BTV-4 strain also detected in 2021 in Corsica (France) (Table 2).

### 3.4. Experimental Infection with the BTV-4 Strain Detected in the Balearic Islands and Comparison with the Widespread BTV-4 Balkan Strain

#### 3.4.1. Clinical Manifestations

All clinical signs detected during the experiment were mild or moderate, with no notable differences between the two groups. None of the non-inoculated animals presented clinical signs of disease nor the presence of a fever.

The onset of a fever (rectal temperature > 40 °C) in the infected sheep occurred between 6 and 9 dpi (Group 1: 5/5 sheep in 6–9 dpi, lasting 1 to 3 days; Group 2: 3/5 sheep in 6–9 dpi, lasting 2–3 days). Comparing both groups, except at 2 dpi (*p* = 0.011), 4 dpi (*p* = 0.036), and 6 dpi (*p* = 0.036), with higher temperatures recorded in group 1, no other statistically significant differences were detected in the course of rectal temperatures (Figure 4).

As for the main clinical findings, mild edema was observed in the neck area from 11 dpi together with serous nasal discharge accompanied by intermittent dyspnoea throughout the experiment in both groups. Other clinical findings, including hyperemic and congestive mucous membranes in the periocular, muzzle, and oral cavity areas, were observed, as well as an increase in the size of retropharyngeal and submandibular lymph nodes. With lower prevalence, ulcers in the oral and nasal mucosa, lameness, weakness, and apathy were also detected in both groups. Ptyalism was only observed in one sheep in Group 2. Table 3 details the clinical signs observed, the count of affected animals, and their onset throughout the experiment.

Only one sheep (Group 1) succumbed to the BTV-4 challenge, being found dead at 12 dpi. This animal experienced fever between 6 and 9 dpi, reaching 41.1 °C on days 6 and 7. The appearance of congestive mucous membranes in the periocular and muzzle area and dyspnoea were identified at 5 and 6 dpi, respectively. Before death, the sheep also showed slight edema in the neck region, lameness, and apathy, observed at 11 dpi.

#### 3.4.2. Macroscopic Lesions

Macroscopic lesions observed in both experimental groups were largely nonspecific. In the two necropsies performed around the peak of viremia, at 7–10 dpi, in one sheep of each group, lesions compatible with bluetongue were found, including hyperemia of the gums (in both sheep), pericardial effusion (Sheep 5, Group 1), and petechial hemorrhages in the papillary muscles of the left ventricle of the heart (Sheep 10, Group 2). Mild pulmonary congestion and enlarged submandibular, retropharyngeal, and mesenteric lymph nodes were also observed in both sheep.

With regard to the two necropsies (one sheep per group) scheduled for 19 and 20 dpi, the only lesions compatible with bluetongue were hemorrhages in the prescapular lymph nodes found in one animal (Sheep 2, Group 1).

At the end of the experiment, at 38–39 dpi, edema at the base of the udder (Sheep 4, Group 1), hydropericardium (Sheep 4, Group 1), and hemorrhages (petechiae and ecchymosis) in the endocardium (Sheep 11, Group 2) were found.

In the necropsy of the only animal that died during the experiment at 12 dpi (Sheep 3, Group 1), a mild subcutaneous edema was found in the submandibular area, together with focal severe subcutaneous hemorrhages (petechiae, ecchymosis, and suffusions) and hematomas at the level of the costal wall near the sternum. Other findings compatible with bluetongue included severe congestion of the ocular mucosa, moderate hyperemia in the gums, severe diffuse alveolar and interstitial edema with the presence of abundant foam in the trachea, serosanguineous nasal discharge, and petechial hemorrhages in papillary muscles of the left ventricle (Figure 5). Moreover, the prescapular, axillary, submandibular, retropharyngeal, tracheobronchial, and mediastinal lymph nodes were enlarged, edematous, and had severe or petechial hemorrhages.

#### 3.4.3. Viral Detection in Blood

The BTV genome started to be detectable in blood samples from infected animals at 3 dpi in both experimental groups. Negative control animals did not show the presence of viral RNA in blood samples throughout the experiment. Statistical differences between both groups were only found (*p* = 0.021) at 3 dpi, with higher levels of viral RNA in Group 2. The peak of RNA detection was observed between 5 and 7 dpi (Ct = 18.9–21.3). Viral RNA levels decayed in both groups after peak detection of up to 20 dpi, then remained constant (Ct = 24.8–28.4) up to the end of the experiment without significant differences (Figure 6). Infectious viruses were successfully recovered from blood at 3 dpi to 13 dpi in Group 1 and up to 10 dpi in Group 2 (Table 4).

#### 3.4.4. Viral Burden in Tissues

The viral genome was detected in all analyzed tissues, showing the pantropic capacity of both viral strains. In all tissues, the highest levels of genomic load were detected at 10 dpi (Group 1) and at 12 dpi (Group 2). Except in heart samples from animals in Group 1 and mesenteric lymph node from animals in Group 2, the viral RNA was detected in all tissues until the end of the experiment (Table 5).

The highest and most consistent levels of viral genome were found in spleen samples throughout the experiment in both groups. Furthermore, as shown in Table 5, the best results in virus isolation were obtained from spleen tissue. An infectious virus was also recovered in lung samples from animals in Group 1 at 7 dpi (Table 5).

#### 3.4.5. Serological Response

All animals were seronegative to BTV at the start of the trial, as well as at 3 dpi. Seroconversion, as determined by ELISA, started at 7 dpi in all infected sheep (Figure 7a). The peak levels of BTV-specific antibodies detected by ELISA were attained between 10 and 13 dpi, remaining elevated until the end of the experiment at 38–39 dpi (Figure 7a). No statistically significant differences were observed between the two groups. The negative control sheep did not seroconvert during the experiment.

Neutralizing antibodies against BTV-4 determined by VNT were also observed from 7 dpi onwards (Figure 7b). The highest titer was recorded at 24 dpi in both groups. As in the ELISA assay, no statistically significant differences were found in VNT titers between both groups.

## 4. Discussion

Bluetongue is a relevant disease of ruminants with high impact in livestock. Over the last two decades, different BTV serotypes have emerged and re-emerged into Europe, causing major disease outbreaks with significant economic consequences. It has been estimated that in Germany alone, the average economic impact between 2006 and 2018 was EUR 180 million [57]. Moreover, the ongoing effects of climatic change favor the expansion of vectors and enhance the transmission of BTV, with devastating outcomes [14,25,58,59]. Due to its severe consequences, the identification of this disease, as well as early detection of incursions of new BTV strains or serotypes, is key to enabling timely and effective implementation of control measures.

Since the first detection of the emerging BTV-4 strain in Majorca in mid-June 2021, the virus spread rapidly throughout the island, eventually expanding to the islands of Ibiza and Minorca. Of note, after the long disease-free period, the animal population was naive to the disease, favoring an epidemic scenario [60], with plenty of new notifications every week until the outbreak was stabilized and controlled. Furthermore, the presence of competent vectors in the Balearic Islands (*Culicoides imicola* and *Culicoides obsoletus*) [61] contributed to the rapid spread of the virus. Similar to other outbreaks caused by BTV in different areas of Southern and Central Europe, the largest number of cases accumulated between late summer and autumn [32,36,38,62]. In this case, the peak of notifications occurred in October, gradually decreasing in the following weeks until the last outbreak was confirmed at the end of November. In addition to the fact that a mass vaccination campaign was launched after the detection of BTV-4, the drop in temperatures below the optimal temperature for the virus replication in vectors (<11–13 °C) [63,64] contributed to controlling the epizootic.

As in other outbreaks caused by BTV-4, the most affected species was sheep, with sporadic cases in cattle and goats [23,36,62]. It should be noted that some of the sheep farms were mixed, with the presence of cattle and/or goats that were not infected or did not report clinical signs of the disease. The variety of clinical signs reported by official veterinary services in the field were typical clinical signs of bluetongue in sheep, which also helped to monitor the epidemic.

The monthly surveillance on sentinel cattle, implemented under the National Surveillance Program for the Control and Eradication for Bluetongue, allowed the detection of BTV-4 circulation even before the appearance of clinical signs in sheep. However, the identification of the serotype involved was initially hampered by the low performance of the method developed and validated specifically for the detection of BTV-4 Western Mediterranean strains that were causing outbreaks in the area in the past. Using a BTV-4 extended rRT-PCR method that included four degenerated positions in the forward primer to amplify Balkan BTV-4 strains solved this problem. Overall, these results suggest that the BTV-4 isolate we dealt with was a different strain, at least in segment 2, from the BTV-4 Western Mediterranean strains. It is worth highlighting the importance of expanding the validation of diagnostic methods with new strains as they emerge to ensure that available recommended methods remain suitable for the detection and identification of new variants. Furthermore, regarding serological diagnosis, the combination of different types of ELISA assays made it possible to increase the sensitivity of the diagnosis, as reflected in the serological diagnosis flow used by the NRL.

The fact that vaccination was implemented during the spread of the disease outbreak allowed for identifying rRT-PCR-positive animals in recently vaccinated farms, as well as seropositive animals with negative rRT-PCR results. Antibody profiles from these two groups were noticeably different, both qualitatively and quantitatively. While a high proportion of infected animals (rRT-PCR-positive) presented seroconversion and high titers of neutralizing antibodies against BTV, a lower proportion of non-infected animals (rRT-PCR-negative) were ELISA-positive, with lower titers of neutralizing antibodies. Further evidence supporting this notion comes from the serological studies performed on the 62 sheep sampled post-vaccination in unaffected areas. Overall, these results suggest a weaker neutralizing antibody response in vaccinated compared to infected animals, which is otherwise expected, given the less immunogenic nature (compared with natural infection) of the inactivated vaccines. However, this should be interpreted with caution, as infection/vaccination status cannot be directly inferred from the tests carried out. Despite these findings, this vaccine has proven useful in the control of this and previous BTV-4 outbreaks. In this regard, some studies with equivalent vaccines have also shown low levels of neutralizing antibodies after one injection, which increased markedly after a booster and prevented viremia and clinical disease [65,66,67]. The level of neutralizing antibodies does not always correlate with the degree of protection after vaccination, and some inactivated vaccines can confer protection in the absence of detectable levels of neutralizing antibodies [12].

The novel isolate BTV-4 SPA (BAL) 2021 shows close sequence identity in the analyzed segments 2, 5, and 10 with BTV-4/21-03 CORSICA 2021, a strain also detected in Corsica (France) in 2021. Furthermore, depending on the segment compared, phylogenetic affinities with different BTV strains can be found, e.g., segment 2 with Balkan BTV-4 strains circulating across the Mediterranean basin since 2014, segment 5 with BTV-4 strains from Spain (2010) and South Africa (2011, 2014), and segment 10 with BTV-1 strains from North African countries (2007). Therefore, this strain, together with BTV-4/21-03 CORSICA 2021, constituted the introduction of a new BTV-4 strain in the Mediterranean region in 2021.

It is worth noticing that the Balkan genotype, with which BTV-4 SPA (BAL) 2021 also presents high homology in segment-2, constituted a new introduction into Europe from North Africa [33,62]. We should also bear in mind that strains of BTV-1, -2, -3, -4, -9, and -16 have entered the Mediterranean basin from North Africa, probably via wind-borne infected midges [36,68,69,70]. Moreover, some of these introductions have spread across the Western Mediterranean islands, as in the case of BTV-2 and BTV-4, which emerged in Sardinia in 2000 and 2003, respectively, and subsequently burst out into the Balearic Islands [69,71]. Based on this evidence, two probable hypotheses can be put forward regarding the origin and route of introduction of the new BTV-4 strain detected in the Balearic Islands: on the one hand, an incursion from North Africa through the dissemination by wind of infected midges; and on the other hand, by the spread of infected vectors from other Mediterranean islands such as Corsica and Sardinia. In this regard, BTV-4 was reported in Tunisia, Corsica, and Sardinia in 2020–2021 [72,73,74]. Further phylogenetic analysis based on Whole Genome Sequencing is highly recommended to fully characterize the molecular epidemiology of this BTV-4 strain.

The pathogenicity and virulence of BTV have been studied extensively for years due to the emergence of different serotypes and strains and their impact on ruminant species. Available evidence indicates that the virulence characteristics of BTV depend on the strain rather than the serotype and are also influenced by the animal host species and breed [75,76,77]. In this regard, we conducted an experimental infection that reproduced the disease caused by two BTV-4 strains, the novel BTV4 SPA (BAL) 2021 and the multi-spread Balkan BTV-4 strain, in a Spanish sheep breed. Low passage virus isolates were used to prepare the inocula in order to avoid attenuation problems already described [77] so as to be able to robustly reproduce the disease in a way most resembling the natural infection observed in the field [78].

The results observed in our experimental infection are consistent with those reported previously in controlled conditions for BTV classical serotypes regarding temperature changes, clinical signs, gross lesions, viremia kinetics (RNAemia), and serological response [5,6,79,80,81,82]. Overall, both strains analyzed here presented similar phenotypic characteristics, with no relevant differences in the onset of clinical signs, viremia, virus tropism, and seroconversion. Both strains produced mild or moderate typical clinical signs of bluetongue [17]. Interestingly, one animal inoculated with the Balkan BTV-4 RNM 2020 strain succumbed to the infection when approaching the viremia peak (12 dpi). This animal presented serous nasal discharge, typical of the disease, in the days prior to death, which became hemorrhagic upon death. This animal exhibited other typical gross lesions in the acute form of bluetongue [17,79,82], some of which were also found in the scheduled necropsies near the viremia peak, as well as at the end of the trial at 38–39 dpi. The long persistence of these alterations (more than one month) could have an impact on the sequelae that sheep develop after infection, even if those only show mild or moderate clinical signs, as in our experimental infection.

We found robust and consistent RNAemia in all inoculated animals, including both strains, which lasted from 3 dpi to the end of the trial (38–39 dpi), peaking at 5–7 dpi. These findings are highly consistent with previous studies, including those made in sheep challenged with other BTV-4 strains [6,82,83]. With regard to the duration of RNAemia, it has been documented that it can extend up to 111–222 dpi [84], being associated with the half-life of ruminant erythrocytes [76], where virions can persist in invaginations of the cell membrane [85,86]. However, infectious viremia is known to be shorter, in a frame from 3 to 54 dpi in sheep [77,82,84]. This aspect has been studied extensively for its relevance in the transmission of the virus, also in cattle [87], and WOAH, in its Terrestrial Code for Bluetongue, established 60 days as the maximum period of BTV infection in ruminants [19]. Our results showed that the detection window of infectious virus for the two BTV-4 strains analyzed did not extend beyond 10–13 dpi. Similar periods of infectious viremia have also been observed in experimental infections in sheep with different BTV serotypes [5,6,80,84] and may depend as well on the BTV serotype and the strain analyzed. Although this transmission period is shorter than that observed for other BTV strains [77], both BTV-4 isolates analyzed here, especially the Balkan BVT-4 strain, which has effectively expanded throughout the Mediterranean basin causing large outbreaks [33,38,39], have shown a great spread capacity under field conditions, suggesting that other factors besides the duration of the infectious viremia play a role in determining the transmission capacity of BTV. Of note, other ruminant species not experimentally analyzed here, especially cattle, despite developing a subclinical infection, may have a greater impact on transmission because they develop longer viremias than sheep [6,77].

The findings in the scheduled necropsies performed during the viremia peak, after the acute phase of the infection, and at the end of the trial were consistent with previous studies on the pathogenesis of BTV [76]. It has been determined that after initial replication in the lymph nodes draining the sites of inoculation, BTV disseminates to secondary replication sites, especially the lungs and spleen, where it replicates in the endothelium and mono-nuclear phagocytes, from which a cell-associated viremia spreads the virions throughout the body [12]. Following that, spleen, lung, and lymph nodes were analyzed in several studies and found to be typical to harbor BTV in infected animals [5,6,88,89]. Our findings are in line with this and highlight the pantropic capacity of both BTV-4 strains. Interestingly, we found consistent levels of BTV RNA in liver and kidney tissues historically overlooked when searching for BTV [81]. These results highlight the findings of Westrich et al. (2024) [81], showing the detection of BTV RNA in those tissues. As expected, the virus was detected to a greater extent in the spleen, where we also succeeded at isolating both virus strains in the first phase of the infection (7–10 dpi). Moreover, the infectious virus was isolated from a lung sample at the peak of viremia. Therefore, although the spleen is still the sample of choice for viral isolation, as recommended by the WOAH [50], our results indicate that other tissues (e.g., lung) from dead animals during the viremia peak could also be suitable for this aim.

Noteworthy, our results show that the strain isolated from the BTV outbreak in the Balearic Islands in 2021 is pathogenic in native Iberian crossbreed sheep. This is relevant given the greater susceptibility described in flocks of European breeds compared to African or Asian breeds [62,75,90]. Moreover, Katsoulos et al. (2016) [62] suggested a possible higher susceptibility of Western European breeds and their crossbreeds to BTV-4 than the indigenous Eastern Mediterranean ones, which could be explained by a virus/host co-evolution in BTV endemic areas [62]. This model allows for extrapolating the findings of the experimental infection described in this study to the field, especially to the areas of the Iberian Peninsula and Southwestern Europe, where breeds related to this type of sheep are raised in livestock.

Historically, it has been reported that the origin of BTV strains emerging in Europe is often linked to Northern Africa [70]. This could be explained, at least partly, by a high global seroprevalence of BTV in ruminants in Africa caused by a wide variety of BTV serotypes [91,92]. Remarkably, the lack of recent available sequences from surrounding Northern African regions hinders the determination of the virus origin or the route of introduction in Europe. Due to the repeated incursions of BTV in the Mediterranean basin, international cooperation for the generation of new sequences of currently circulating strains is highly necessary.

Bluetongue remains one of the current challenges in animal health due to the continuous emergence of new pathogenic strains and serotypes with the potential to cause major outbreaks with impact on the livestock health safety. In this context, early diagnosis and viral characterization are crucial for implementing control measures. Here, we show that the index case in the Balearic Island 2021 outbreak was an asymptomatic sentinel bovine subjected to active surveillance, from which control efforts were raised. Through experimental infection, we confirmed the pathogenic potential of the emerging BTV-4 strain detected, which exhibits phenotypic characteristics analogous to the disseminated Balkan BTV-4 strain. Therefore, this study illustrates the key role of active surveillance, both serological and virological, in the early detection of virus circulation in disease-free areas and in the identification of novel viral variants. In this regard, international cooperation is becoming increasingly important for the implementation of effective surveillance and outbreak management strategies, as well as for scientific research in this field.

## Figures and Tables

**Figure 1 microorganisms-13-00411-f001:**
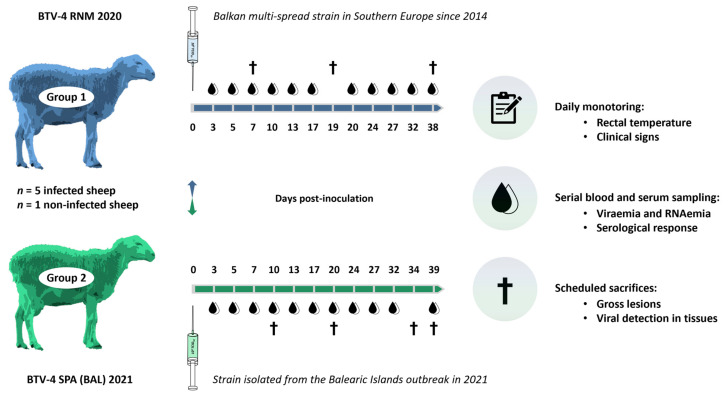
Experimental infection study design.

**Figure 2 microorganisms-13-00411-f002:**
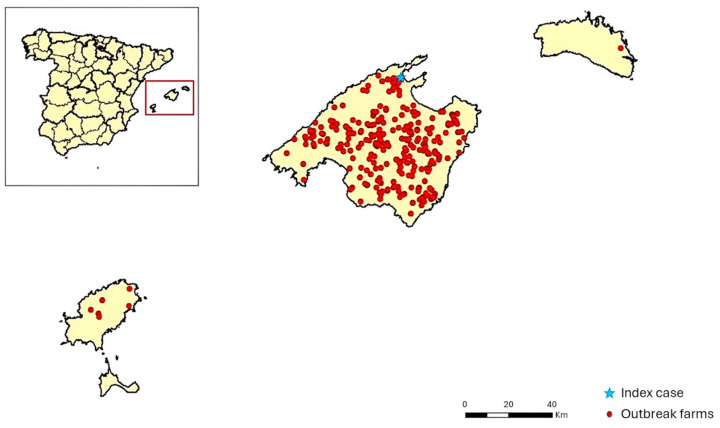
Map of confirmed outbreaks during the BTV-4 epidemic in the Balearic Islands in 2021. The red circles represent the location of each infected farm, and the index case is indicated with the blue star. Source: Ministry of Agriculture, Fisheries and Food (modified).

**Figure 3 microorganisms-13-00411-f003:**
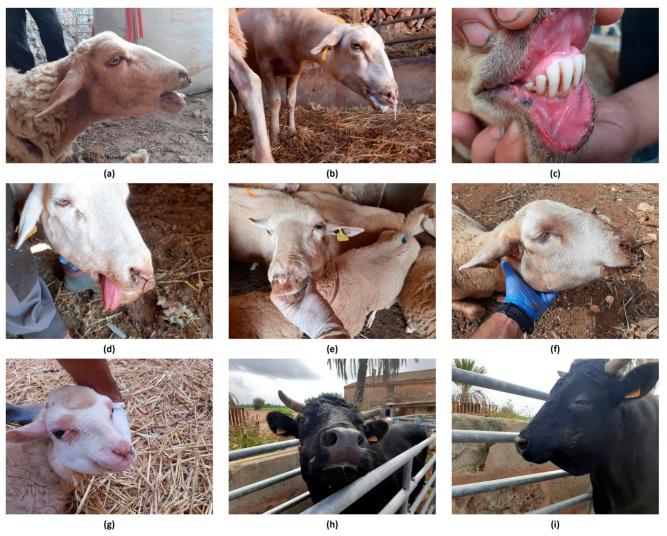
Clinical signs observed in BTV-4-infected animals during the outbreak in the Balearic Islands in 2021. (**a**): orthopneic position, facial edema, serous nasal discharge; (**b**): orthopneic position, ptyalism with the presence of foam in the mouth and nasal discharge; (**c**): congestion of the oral mucosa, ulcers, and inflamed gums; (**d**): glossitis and serous nasal discharge; (**e**): nasal discharge; (**f**): facial and submandibular edema and serous nasal discharge; (**g**): hyperemic mucous membranes in the periocular and muzzle area, serous nasal secretion, and facial and submandibular edema; (**h**) ptyalism; (**i**) facial and submandibular edema. Images taken during the outbreak and provided by the Official Veterinary Services of the Balearic Islands.

**Figure 4 microorganisms-13-00411-f004:**
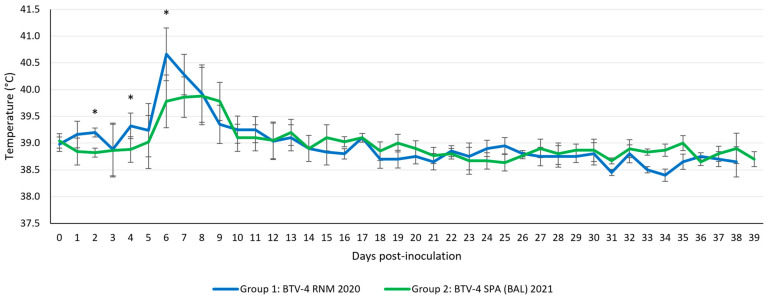
Temperature records in both groups. Error bars show the standard deviation of the data. * *p* value < 0.05.

**Figure 5 microorganisms-13-00411-f005:**
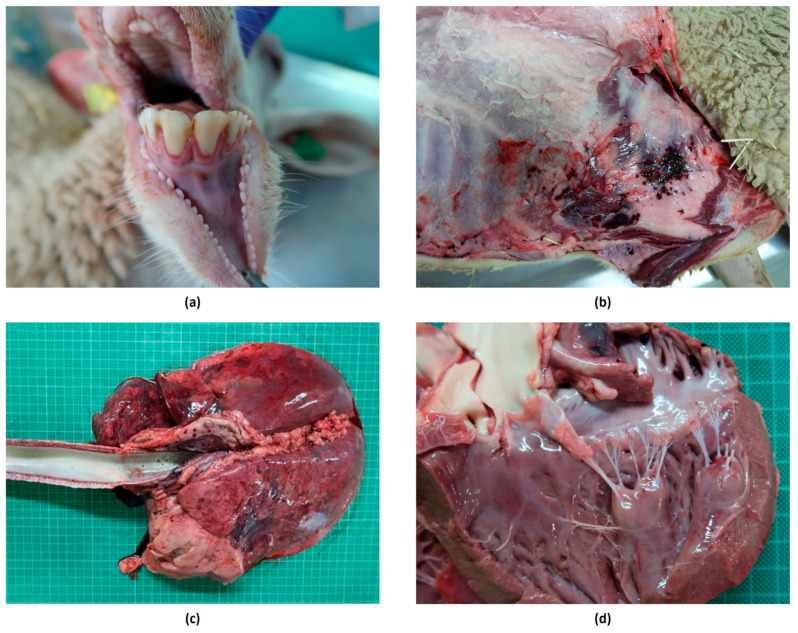
Macroscopic lesions observed in the sheep that succumbed to BTV infection in Group 1. (**a**): hyperemia in gums; (**b**): hemorrhages and subcutaneous hematomas; (**c**): lungs with congestion, hemorrhages, alveolar and interstitial edema, and presence of foam in the trachea; (**d**): petechial hemorrhages in the papillary muscles of the left ventricle.

**Figure 6 microorganisms-13-00411-f006:**
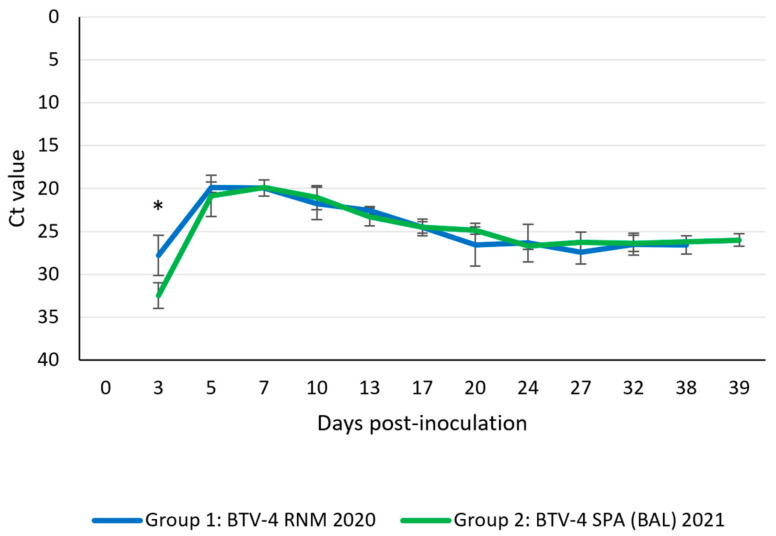
Viral ARN load in blood samples. Error bars show the standard deviation of the data. * *p* value < 0.05.

**Figure 7 microorganisms-13-00411-f007:**
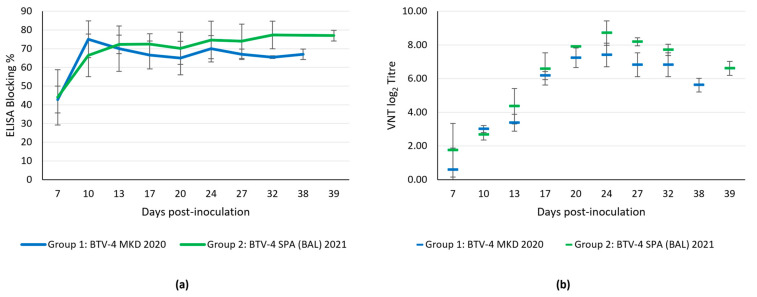
Serological response in both experimental groups. (**a**) BTV serogroup antibody determination by ELISA. (**b**) Neutralizing BTV-4 antibodies determined using VNT. Antibody titer by VNT is expressed as the log_2_ of the titer obtained using the Spearman–Karber method. Error bars show the standard deviation of the data in both graphs.

**Table 1 microorganisms-13-00411-t001:** Primers and probes of serotype-specific rRT-PCR for BTV-1, -4, and -8.

Serotype/Topotype	Primers (F/R) and Probe (P)	Sequence 5′-3′	Product Size (pb)	Validated to Detect
BTV-1 w.med	BTV-1 w.med (F)	GACGATGCCGCGTATGGT	135	Western Mediterranean strains
	BTV-1 w.med (R)	TTAGCTTTGCATCCTTTTCAAAA		
	BTV-1 w.med (P)	FAM-TAGAACGTTACCTTCTGATTTT-MGB		
BTV-8 eur	BTV-8 eur (F)	TGCGCACGATATYCGAATT	66	European strains
	BTV-8 eur (R)	GACGTCAGCCCAAAACGATT		
	BTV-8 eur (P)	FAM-TTGTACGCTCCAACYTCCAAAGGTA-BHQ1		
BTV-4 w.med	BTV-4 w.med (F)	AACACGTATTTATTGTCCTCCAATTG	59	Western Mediterranean strains
	BTV-4 w.med (R)	AGCTTGCGGCCGGAAT		
	BTV-4 w.med (P)	FAM-CGTTCCCGTTGACCG-MGB		
BTV-4 ext	BTV-4 ext (F)	AACACRTAYTTATTGTCYTCYAATTG	59	Western Mediterranean and Balkan strains
	BTV-4 ext (R)	AGCTTGCGGCCGGAAT		
	BTV-4 ext (P)	FAM-CGTTCCCGTTGACCG-MGB		

w.med: Western Mediterranean; eur: European; ext: extended; Primer (F): forward primer; Primer (R): reverse primer; pb: pair base. All primer sequences are developed for targeting Segment 2 (VP2 structural protein).

**Table 2 microorganisms-13-00411-t002:** The percent of nucleotide sequence identity for segments 2, 5, and 10 of BTV-4 SPA (BAL) 2021, with the top similarity matching BTV sequences in GenBank.

Viral Segment Material ID (AN)	Length (pb)	nt Position in BTV-4/GRE2014/08 (AN)	%	Strain	AN
Partial seg-2 isolate BTV-4 SPA 2021/01 VP2 (MZ919337.1)	1126	767 to 1892 (MT879212.1)	99.73	BTV-4/21-03 CORSICA 2021	PP262563.1
			98.05	BTV-4/16-03 CORSICA 2016	KY654329.1
			98.05	BTV-4/KOS2014/01	OP186417.1
			98.05	BTV-4/GRE2014/08	MT879212.1
Partial seg-5 isolate BTV-4 SPA 2021/01 NS1 (MZ919338.1)	260	22 to 281 (MT879215.1)	100	BTV-4/21-03 CORSICA 2021	PP262566.1
			98.85	BTV 17/O.aries-tc/ZAF/2014	MG255486.1
			98.85	BTV 5/O.aries-tc/ZAF/2011	MG255456.1
			98.85	BTV-4 isolate SPA2010/01	KP821432.1
Partial seg-10 isolate BTV-4 SPA 2021/01 NS3 (MZ919339.1)	760	3 to 762 (MT879220.1)	99.87	BTV-4/21-03 CORSICA 2021	PP262571.1
			98.55	BTV-1 TUN2007/01	KP821975.1
			98.55	BTV-1 MOR2007/01	KP821975.1
			98.55	BTV-1 LIB2007/06	KP821973.1

%: percent nucleotide identity; nt: nucleotide; pb: pair base; AN: GenBank Accession Number.

**Table 3 microorganisms-13-00411-t003:** Clinical signs observed throughout the trial.

Clinical Sings	Group 1		Group 2	
	Onset dpi	*n*	Onset dpi	*n*
Hyperemic and congestive mucous membranes	5–6	5/5	3–13	5/5
Fever	6–9	5/5	6–9	3/5
Serous nasal discharge	5–6	4/5	5–7	4/5
Dyspnoea	6	4/5	6–24	3/5
Increase in size on retropharyngeal and submandibular lymph nodes	6–28	2/5	7–17	4/5
Neck edema	11–20	3/4	11–24	3/4
Weakness and apathy	11–28	2/4	12–26	2/4
Muzzle edema	18	1/3	17	2/4
Lameness	11	1/4	11	2/4
Ulcers in the oral and nasal mucosa	17	1/3	11	1/4
Ptyalism		0/5	3	1/5

Onset dpi: day post-inoculation on which the appearance of the clinical sign is first observed; *n*: number of sheep showing the clinical sign out of the total when each first clinical manifestation was reported. Due to the scheduled sacrifices, the number of animals during the experiment was not constant, so the appearance and the development of clinical signs were not always assessed on a total of 5 sheep.

**Table 4 microorganisms-13-00411-t004:** Virus isolation in blood during the experiment.

Virus Isolation in Blood (Positive or Negative; Number of Positive Animals/Total Animals)											
Group 1	3 dpi	5 dpi	7 dpi	10 dpi	13 dpi	17 dpi	20 dpi	24 dpi	27 dpi	32 dpi	38 dpi
	+; 1/5	+; 5/5	+; 4/5	+; 3/4	+; 1/3	−; 0/3	−; 0/2	−; 0/2	−; 0/2	−; 0/2	−; 0/2
Group 2	3 dpi	5 dpi	7 dpi	10 dpi	13 dpi	17 dpi	20 dpi	24 dpi	27 dpi	32 dpi	39 dpi
	+; 1/5	+; 5/5	+; 5/5	+; 4/5	−; 0/4	−; 0/4	−; 0/3	−; 0/3	−; 0/3	−; 0/3	−; 0/2

dpi: days post-inoculation. (+): positive (at least one animal is positive); (−): negative (all animals are negative)

**Table 5 microorganisms-13-00411-t005:** Virus detection in tissue samples during the experiment.

Group	Dpi	Liver	Lung	Heart	Spleen	Kidney	Mes. l.n.	Med. l.n.
Group 1	7	+++	+ *	+++	+++ *	++	++	++
	12	+++	+++	+++	+++	++	++	++
	19	++	++	+	+++	++	+	+
	38	+	+	−	++	+	+	+
Group 2	10	+++	+++	+++	+++ *	+++	+++	++
	20	++	++	+	+++	+	++	++
	34	++	+	+	++	+	+	++
	39	+	+	+	++	+	−	+++

+++ = 15–25 Ct; ++ = 25.1–30 Ct; + 30.1–35 Ct; − = Ct ≥ 35. * Virus isolation achieved (attempted in liver, lung, heart, spleen, and kidney). dpi: days post-inoculation; Mes. l.n.: mesenteric lymph node; Med. l.n.: mediastinal lymph node. The tissue samples were obtained from a single animal (*n* = 1) with the exception at 38 dpi in Group 1 and at 39 dpi in Group 2, which comprise two sheep (*n* = 2, expressed as the average value).

## Data Availability

The original contributions presented in this study are included in the article. Further inquiries can be directed to the corresponding author.

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
