# Peer review of "Emerging Bluetongue Virus Serotype 4 in the Balearic Islands, Spain (2021): Outbreak Investigations and Experimental Infection in Sheep"

_microorganisms, 2025, doi:10.3390/microorganisms13020411_

Round 1

Reviewer 1 Report

Comments and Suggestions for Authors

Recently, I reviewed a study entitled Recently, I reviewed a study entitled “ Emerging Bluetongue Serotype 4 strain in the Balearic Islands, 2021: Outbreak Investigations and Biological Characterization in Sheep,” presented by David Romero-Trancón, Marta Valero-Lorenzo, María José Ruano, Paloma Fernández-Pacheco, Elena García- Villacieros, Cristina Tena-Tomás, Ana López-Herranz, Jorge Morales, Bartolomé Martí, Miguel Angel Jiménez-Clavero, Germán Cáceres, Montserrat Agüero and Rubén Villalba. Partial sequencing of the BTV-4 strain identified in several Balkan islands, outbreak caused by the BTV-4 in 2021, diagnostic methods and experimental infection of sheep were presented in this manuscript.

All over the text change suggestions:

Bluetongue as disease: I suggest to use abbreviation (BT) after the first its mention in the text.

serogroup specific (GS)- readers more acquainted with “pan-BTV RT-qPCR”. I suggest to exchange “GS” with “pan-BTV RT-qPCR”. TS- to specific serotype. For example, “BTV-4 specific RT-qPCR”.

Results: phylogenetic tries of all three segments should be also presented.

Reference in the text doesn’t meet requirements of the journal.

Title: “Biological Characterization in Sheep” it should be re-phrased. What was characterized: virus, clinical manifestations of the disease, something else? To clear

Abstract.

Lines 20-23. “In this study we describe the BTV-4 outbreak in the Balearic Islands in 2021 including clinical and laboratory findings as well as a preliminary molecular characterization of the strain involved by partial sequencing of segments 2, 5 and 10 of the viral genome.” I suggest to divide the sentence for several sentences. Its current form looks non-professional.

Line 23. “in vivo characterization by experimental infection”. In vivo characterization and experimental infection is almost the same.

Line 24. “a known widespread BTV-4 strain”. For the reader is not “known”. Specific information about the” known widespread BTV-4 strain” has to be added.

Lines 23-25. “Additional in vivo characterization by experimental infection in sheep was carried out in parallel with a known widespread BTV-4 strain to determine and compare their virulence and pathogenicity, as well as the course of the infection and asses their laboratory diagnostic parameters.” The sentence is too long that complicate the reading and understanding the information. I suggest to divide it.  

Lines “pathogenic capacity” and “diagnostic parameters” is not properly used terminology. To re-phrase.

Keywords: bluetongue- to add virus, BTV-4- to delete, diagnoses- to change with “diagnosis”. I suggest to add additional keywords

Introduction.

I suggest to add information on BTV-4 in such Mediterranean countries as Israel and Turkey.

Line 33. “known as Bluetongue” to delete. It is a rare clinical sign already.

Line 34. “and less frequently” of what? It should be written clear.

Lines 36-39. The sentence should be re-written.

Line 70. “epidemic” to exchange with epizootic.

Line 74. BTV-3 (2023) in Netherlands – it is not present only Netherlands. It begins from Morocco, Italy and Israel, recently it registered in Germany, England, Italy, Belgium, France, Denmark, Norway, Sweden, Portugal. This data has been proved.

Lines 72-76. I suggest to divide the sentence into the several sentences.

Lines 111-124. This information is related to the next section/s. It should be seriously shortened.

Lines 128-129. “the most severely affected species during the outbreak”- it is redundant. To delete.

Lines 125-132. The aims of the study are not clear. They should be written strait.

Materials and methods.

Line 135. “2.1.1. Clinical examination and sampling by the Official Veterinary Services” – the title of the sub-section. I didn’t find any clinical examination in the sub-section. The title should be proved.

Lines 142-147. In the sentence the word “to evaluate” was used twice. To correct.

Line 143. “after vaccination” with which king of vaccine against what pathogen?

Line 148. “Further ethical approval was, therefore, not needed.” It is not the place for this information. It should be moved to the “Ethical approval:”

Lines 149-155. “2.1.2. Serological test flow chart.” This information is useless and should be united in the sections below describing ELISA and VNT.

Lines 157-164. “2.1.3. Virological diagnosis flow chart”. This information is useless and should be united in the sections below describing PCR.

Line 165. “2.2. Biological characterization by experimental infection in sheep” . I suggest to delete  Biological characterization” and add with which viruses it was done.

Lines 167-177. This paragraph should be written clearer. Too much useless phrase and information.

Line 188. “TS rRT-PCR” Authors tested for BTV-4 specific RT-qPCR, didn’t them? If so, the exact data should be written.

Line 192.  Subsection 2.2.3. Sampling and clinical monitoring of the animals”. The section should include things which presented in the title. Sample storage, preparation, ELISA, PCR, virus isolation- should be moved for specific sections/subsections.

Lines 193-195. “During the experiment, animals were monitored daily by veterinarians to detect the

onset of clinical sings compatible with Bluetongue and to record rectal temperatures (considered as fever when >40ᵒC) until day 39 post-inoculation (dpi).” It should be shortened. The reader is not a child. I underlined words, which can be deleted without damage of the text.

Lines 195-198. “Blood samples (with and without EDTA) to monitor RT-PCRemia, viraemia and serological response were obtained on days 3, 5, 7, 10, 13, 17, 20, 24, 27, 32 and 38-39 dpi and randomly selected animals (one per group) were sequentially sacrificed at different dpi as indicated in Figure 1 to determine the presence of gross lesions and virus tropism.’ “Blood samples (with and without EDTA) to monitor RT-PCRemia, viraemia and serological response”- all this can be shortened up to “whole blood and serum samples were collected according to scheme presented in the Figure 1”. Regarding euthanasia- to change in the same way.

Lines 201-204. “For the study of the serological response, after obtaining blood samples in vials without anticoagulants, the samples were stored at 37°C for 1 hour for coagulation and then

left at 4°C overnight. Samples were then centrifuged at 2400 g for 5 min, and the sera was collected and stored at –20°C until analysis.” I suggest to delete it. It is well known data for 1st year students.

 Lines 204-207. “The sera were analysed to detect the presence of antibodies against the serogroup-specific VP7 protein using the b-ELISA kit INGEZIM BTV Compac 2.0 (Ingenasa, Madrid, Spain). In addition, the presence of neutralising antibodies against BTV-4 was analysed by VNT in ELISA positive sera.” To move to ELISA section, 2.3.1. VNT- move to VNT.

Lines 208-209. “For the virological study, EDTA blood were kept refrigerated at 4°C after being obtained until analysed for GS rRT-PCR targeted to segment 10 of the viral genome.” To delete. It is 1. Repetition. 2. Well known things on BTV storage.

Line 216. To delete.

Lines 217-222.  2.2.4. Ethics statement. Is should be moved to “Ethical approval:”

Lines 240-245. The smallest titer while the serum was evaluated as positive, should be marked. No data about the used strain/strains for VNT is presented. This information has to be added to the text.

Lines 259-260.The primers, probes, amplicon sizes and strains for which these methods were validated are shown in Table 1.

Table 1. Primers and probes of serotype specific rRT-PCR for typing: BTV-1 (western mediterranean

strain); BTV-8 (European strain); BTV-4 w.med (western Mediterranean strain); and BTV-4 extended

(western Mediterranean and Balkan strains) targeting Segment 2 (VP2 structural protein). According the role of the journal, the title of a table should include the title without any explanation. All additional information has to be moved to the footling of the table. All presented abbreviations should be completed by full names in footling of the table. The column “Product size (pb) - the title of the column includes “bp”- so, the length of the product should show numbers only.

Lines 265-271.  Not clear at all. I suggest or re-write, or present information in a table.

Lines 282-283. “Next, the erythrocytes were lysed by osmotic shock by adding 1 ml of sterile distilled water and maintaining the vial in ice for 10 min.” The authors forgot to show proportion of whole blood cells to water.

Line 308. Title: 2.3.6. Partial sequencing of BTV-4 segments-2, -5 and -10.” Lines 309-318- there are information on PCR, gel/PCR purification and analyzing of the data. The title should be corrected according to presented information. I suggest to change the title to “Sequencing and sequencing analysis”

Results.

Line 320. “evolution of the outbreak” it should be re-phrased.

Line 333. “three other cattle farms and one goat farm,” the thought is uncompleted. To complete the phrase/sentence.

Line 335. “1.5% mortality).” It is spoke on proportion to total number of animals in the farm? Please, explain and complete the data.

Line 337. “4.8% mortality).” The same.

Figure 3. the footling in the figure should present maximum information in minimum words. All information has to be present in the same way:  The authors present information at the beginning ”(A):information;“ it should be done till the last picture/letter (i).  the presentation of the picture doesn’t meet request of the journal. It has to be changed

Line 373. “3.3. Laboratory diagnostic activities during the outbreak”. I suggest to change the title of the section to “3.3. Laboratory diagnosis” and subdivide the section to several sub-sections according to the types of the tests.

Lines 374-383. The section should be simplified and shortened. It is not convenient for reading.

Lines 387-390. Not clear. The sentence should be subdivided into several sentences.

Lines 393-414. I suggest to show this data in the table. It in present version it is unreadable.

Line 415. Phylogenetic tries of all three segments should be also presented.

Line 421. Title of the table should be re-written.

Lines 423-424. “3.5. Biological characterization of the BTV-4 strain detected in the Balearic Islands and comparison with the widespread BTV-4 Balkan strain” .The are results of the experimental infection. Th title has to include words of “experimental infection”.

Line 425.  3.5.1. Clinical observations”. It is “clinical manifestation”. To change

Lines 431. “Except at 2 dpi (p = 0.011), 4 dpi (p = 0.036) and 6 dpi (p = 0.036)”. it is not understood, what it is mean. Whether is it comparison mean temperature between the groups? The information has to be added.

Lines 442-443. “Table 3 indicates the clinical signs identified, the number of affected animals, and their onset during the experiment timeline.” The word “details” has to be added to the sentence. The sentence has to be re-written. Table 3 should be moved to the first place where the Table 3 was mentioned. The same with Figure 4.

Table 3. Title should be changed.

Line 455. “It should be noted that” I suggest to delete it.

Table 4. table doesn’t meet the request of the journal. “+” and “_” are redundant. It is understood.

Footling – “(+): successful virus isolation; (-): negative virus isolation in”- to delete.

Line 504 “tissues analysed” to rewrite

Line 518. “except at 38 dpi in Group 1 and 39 dpi in Group 2 that come from 2 sheep (n=2, expressed as average value).

Lines 521-522. “Seroconversion, as determined by ELISA, started at 7 dpi in all infected sheep”. Which days post infection before 7dpi were also tested? This information has to be added to the text.

Lines 522-523. “High and constant levels of BTV- specific antibodies were observed from 10 dpi”. Considering presented data, I disagree with the statement. To change.

Lines 526-527. “Neutralizing antibodies against BTV-4 determined by VNT were observed also from 7 dpi onwards”.  The same

Discussion.

Lines 550-567. Some data was already presented in the “introduction” and “results” section. It should be avoided.

Lines 592-599. These results already were presented in the “results” section. To delete.

Lines 589-612. It is clear what the main issue of the paragraph. It should be shortened and re-written.

Lines 621-622.  It is worth noticing that the Balkan genotype, despite grouping with the Western topotype and presenting genetic proximity with the segment-2.” Since there is no data on phylogenetic analysis, it is impossible to classify the strain.

Lines 636-638.” Further phylogenetic analysis based on Whole Genome Sequencing is being performed to fully characterize the molecular epidemiology of this BTV-4 strain.” Reference should be added.

Lines 724 -726. “the origin of strains emerging in Europe is often linked to Northern Africa, where a high global seroprevalence” How this two facts: North African origin of the BTV strains and global seroprevalence? Explain and correct

Line 741. “Illustrates” change the capital to the small letter of the word.

 Supplementary data cannot be opened. 

Comments on the Quality of English Language

Quality of English is weak

Author Response

Response to the Reviewer 1 comments

Thank you very much for taking the time to review this manuscript. We agree with most of your comments and it has been very useful to significantly improve the manuscript.

Please find the detailed responses in the document attached.

Sorry for the inconvenience to download the supplementary material. It has been uploaded this time in a different format.

Reviewer 2 Report

Comments and Suggestions for Authors

Emerging Bluetongue Serotype 4 strain in the Balearic Islands, 22021: Outbreak Investigations and Biological Characterization in Sheep. An outbreak of Bluetongue virus (BTV-4) in the Balearic Islands and its biological characteristics. The design of the research method is rigorous, the data analysis is reasonable. In the selection of diagnostic methods, sample processing, virus detection and data analysis have shown a high scientific and normative.This manuscript has reference value for the epidemic and prevention of Bluetongue virus, but there are many shortcomings.

1. The introduction details the Bluetongue virus (BTV) and its impact on livestock, clarifying the history and severity of the spread of BTV in Europe.It is recommended to refine the introduction and not repeat the discussion section.

2. 125-132:The purpose and significance of the study of this manuscript are also introduced.

3. Modify Table 1 according to the format of this publication.

4.Result: Paragraph 3.3, the description should be brief, clear and concise.

5.Further improve the quality of figures 2,6,7.

6. Table 2,3,4,5 Quality needs improvement, Table 3 part, it is recommended to add statistical processing.

7.If the experiment permits, it is recommended to increase the number of pathological tissue sections and immunohistochemical tests.

8. It is recommended to rewrite the summary.

9.The reference part is messy, it is suggested to rewrite the format of this journal.

10.Added full text sulk handling.

Author Response

Response to the Reviewer 2 comments

Thank you very much for taking the time to review this manuscript. We agree with most of your comments and it has been very useful to improve the manuscript.

Please find the detailed responses in the document attached.

Reviewer 3 Report

Comments and Suggestions for Authors

A very interesting investigation. It was a pleasure to read the description of such a competent investigation of an epidemiological outbreak. It is a bit of a pity that the genetic part is a bit lame (only one genome has been sequenced, and only partially). With modern sequencing capabilities, one would expect that in such a full-fledged, large-scale work, the complete genomes of the viruses would also be determined. But still, very good work has been done. And well, clearly presented.

Minor remarks:

Line 64 "naïve temperate regions"

Please, reformulate this sentence. What is it, "naïve regions"?

Line 68 "Until 2006, BTV was considered an exotic pathogen of tropical and subtropical regions"

Could you please, to clarify this assumption. Why was it "exotic pathogen of tropical and subtropical regions" if "from 1998 to 2006, BTV-1, -2, -4, -9 and -16 spread through out the European region of the _Mediterranean_coast "?

Line 333 "Except for the index case"

What is the "index case"?

Line 553

"population was naïve"

Do you mean "naive", not "naïve"?

Line 637

"is being performed"

"Should be performed in future", you mean? Or "currently in progress in our labs"? 

You have not done a full genome analysis in this work, only a partial one.

Author Response

Response to the Reviewer 3 comments

Thank you very much for taking the time to review this manuscript. We agree about whole sequencing. Please find the detailed responses in the document attached.

Round 2

Reviewer 2 Report

Comments and Suggestions for Authors

Emerging Bluetongue Serotype 4 strain in the Balearic Islands, 22021: Outbreak Investigations and Biological Characterization in Sheep. An outbreak of Bluetongue virus (BTV-4) in the Balearic Islands and its biological characteristics. The design of the research method is rigorous, the data analysis is reasonable. In the selection of diagnostic methods, sample processing, virus detection and data analysis have shown a high scientific and normative. This manuscript has reference value for the epidemic and prevention of Bluetongue virus, The author has made a lot of changes to the question raised and proposes to publish them after minor revisions.

1. 212:to monitor RT-PCRemia ,Whether there are spelling errors?

2.Suggestion 2.3.6 Subheading rewrite.

3.Suggest checking full text to modify punctuation to match and do full text sulk.

Author Response

Thank you for your time to review the manuscript again

  1. 212:to monitor RT-PCRemia ,Whether there are spelling errors? We have maintain the word RT-PCRemia but we have changed the sentece to explain their meaning which has been used in a Figure. Final sentece: "to assess the presence of BTV genome (RT-PCRemia), infectious virus (viraemia) and sero-logical response"

2.Suggestion 2.3.6 Subheading rewrite. "Partial genome sequencing and sequencing analysis"  to mention which sequencig performed was partial

3.Suggest checking full text to modify punctuation to match and do full text sulk. A review has been made of everything, correcting some details. The names of the supplementary tables have been changed (Table S1, S2, S3 and S4). Figure 2 has been moved to where it was mentioned.
